# LLMAD-mini: Efficient Distilling Hierarchical Chain-of-Thought for Interpretable Log Anomaly Reasoning and Detection using Large Language Model

## Abstract

Log anomaly detection is critical for system reliability, yet most existing methods focus only on binary detection without providing explanations or identifying root causes, which limits their usefulness in production environments. To address these challenges, we propose LLMAD-mini, a lightweight LLM-based model that combines knowledge distillation with Low-Rank Adaptation (LoRA) fine-tuning to deliver strong reasoning and comprehensive log understanding. Large language models (LLMs) with human-interpretable descriptions can be tuned for specialized logs via supervised fine-tuning, but the high cost of training and deployment remains a major barrier. To achieve efficient adaption on small in-domain dataset on LLMs, we introduce a hierarchical Chain-of-Thought mechanism that significantly enhances reasoning capability with limited data. Evaluated on different system log datasets, LLMAD-mini surpasses traditional anomaly detection methods in detection accuracy and provides far better reasoning than much larger LLMs. Notably, it achieves a 3.2× improvement on reasoning quality compared to a LLM with 30× more parameters. Furthermore, our experiments on out-of-domain logs demonstrate LLMAD-mini's ability to generalize across diverse systems with the improvement of 40% of accuracy on anomaly detection and improve the Bleu-4 from 0.01 to 0.49 while diagnosing failures, making it a practical and efficient solution for real-world deployment.

## 1 Introduction

Software systems are fundamental for the operation of modern infrastructure. Nowadays, these systems are characterized by significant complexity, distributed architectures, and massive scales, which makes them powerful but also fragile. Yadav et al. (2020); Meena Siwach (2022) This complexity inherently poses challenges in ensuring reliability of systems. Consequently, the occurrence of system anomalies—unexpected behaviors are inevitable. These anomalies can trigger serious problems in software systems which can lead to reduced system performance and corrupted data and further causes substantial financial losses. Pang et al. (2021) Therefore, it is naturally raised the demand about effective anomaly detection methods on complicated systems to maintain the overall health of software infrastructure.

Engineers commonly rely on system logs to manage the status of running systems. These logs are enriched with detailed information about log events with timestamps.Chalapathy & Chawla (2019) In theory, such data of log events inherently contain core patterns to understanding system behavior and diagnosing problems. The challenge, however, lies in the huge volume of this data for human to read, which makes the development of automated log-based anomaly detection methods be much more essential. Wei et al. (2024)

To solve this problem, log anomaly detection methods have been proposed using traditional machine learning and later, deep learning models like LSTMs Hochreiter & Schmidhuber (1997) and Transformers Vaswani et al. (2017). Moreover, those models have shown a competitive performance on binary classification task on identifying anomaly/non-anomaly logs.Yadav et al. (2020);

Meena Siwach (2022) Whereas they achieve effecient at detection, what information engineers obtain from those methods is that an anomaly has occurred or not in given logs, without any relevant reasoning or a human-understandable explanation for the decision. This lack of interpretability is a severe obstacle for engineers for digging out the root cause or dealing with the issue. In addition, existing methods suffer from a critical limitation of generalization, as they are only applied to the identical system logs and environment on which they were trained. This makes them inherently non-portable across different systems and vulnerable to obsolescence from any log format change or system update. Consequently, these models are inflexible and costly to maintain in evolving, real-world infrastructures.

The recent explosion of Large Language Models (LLMs) Zhao et al. (2023) presents a new and exciting frontier on text generation. LLMs shows a remarkable performance on general language and semantic understanding Minaee et al. (2024); Naveed et al. (2025). It is likely that LLMs can provide explanations and root cause analysis on system logs, but may lack of enough knowledge on domain-specific area, like log anomaly analysis. Furthermore, deploying these massive LLM models for log monitoring is often impractical, their huge model size, high computational cost and hardware requirements, significant inference latency, and reliance on API access lead them to be unrealistic for deployment on real production environments.

In this paper, we introduce a novel method LLMAD-mini, which is designed to resolve the previous challenges presented in both traditional methods and LLMs. To achieve this, we trained our model with knowledge distillation mechanism Hinton et al. (2015); Xu et al. (2024) by transferring the advanced reasoning abilities of a large "teacher" LLM to our model. The core idea is to adopt our novel hierarchical Chain-of-Thought (CoT) Wei et al. (2022) to elicit step-by-step reasoning from a large LLM on why a specific log sequence is anomalous. We then use this generated reasoning to fine-tune our model. The fine-tuned student model learns not just to classify logs as normal or anomalous but to perform the reasoning process itself, further provide log analysis and possible error cause with engineers. Above all, our key contributions are as follows:

- We introduce a hierarchical Chain-of-Thought mechanism including event-wise CoT, stage-wise CoT, pattern CoT and indicator CoT that achieves 3.2× higher Bleu-4 scores than models 30× larger, significantly enhancing reasoning capabilities through structured knowledge distillation.

- Through comprehensive experiments, we show that LLMAD-mini achieves 0.97 F1-score on anomaly detection, surpassing traditional log analysis methods while providing interpretable explanations.

- Our approach enables practical deployment with minimal computational overhead, requiring only 2 hours of training on a single GPU through efficient LoRA-based knowledge distillation.

- We demonstrate strong generalization to unseen domains, achieving 0.72 F1-score and 21.5× higher Rouge-2 scores than baselines on out-of-domain HDFS logs without additional training.

## 2 RELATED WORK

**Log Anomaly Detection**: Many traditional methods, which formulate log anomaly detection problem as binary classification task, have been proposed since 2017. Some methods including Du et al. (2017); Meng et al. (2019); Zhang et al. (2021); Catillo et al. (2022); Xie & Yang (2023); Zhang et al. (2023b); Duan et al. (2021); He et al. (2023) employs LSTM, Transformer, GAN Goodfellow et al. (2020) or autoencoderLeCun (1987) as main framework to predict the next most possible normal log event from previous log sequences, treating unexpected event occurred as anomalies. Other methods such as Lu et al. (2018); Zhang et al. (2019); Yang et al. (2021); Zhao et al. (2022); Xie et al. (2022); Zhang et al. (2023a); Hashemi & Mäntylä (2024) trained the model based on CNN LeCun & Bengio (1998), GNN Scarselli et al. (2008), transformer as binary classifiers to determine if a given log sequence is normal or abnormal. More recently, with the remarkable success has been achieved by LLMs, works like Qi et al. (2023); Egersdoerfer et al. (2023); Liu et al. (2024e); Pan et al. (2024) leverage prompt engineering without any tuning on LLMs to detect anomalies directly in terms of the pre-trained knowledge while Guo et al. (2021); Lee et al. (2023); Lin et al. (2024);

Almodovar et al. (2024); Chen & Liao (2022); Jilcha et al. (2024); Hadadi et al. (2024); Guan et al. (2024) fine-tuned LLMs to achieve more accurate detection performance by adapting on specified datasets.

**Large Language Model**:The landscape of large language models has evolved through several foundational families. GPT-3 Brown et al. (2020) demonstrated that 175B-parameter models could perform diverse NLP tasks, while GPT-4 Achiam et al. (2023) advanced multimodal capabilities and reasoning benchmarks. Gemini Team et al. (2023) unified text, image, audio, and video processing, with Gemini 1.5 Pro supporting up to 1 million tokens. The LLaMA family Touvron et al. (2023); Dubey et al. (2024) proved smaller models trained on more data can outperform larger ones, revolutionizing open-source development. DeepSeek Liu et al. (2024b;c); Guo et al. (2025) advanced mixture-of-experts architectures for improved parameter efficiency. The Qwen family Bai et al. (2023); Team (2024) evolved from Llama-based architectures to Qwen2.5 with 3B-72B parameter variants, while Qwen3 Yang et al. (2025) introduced hybrid thinking modes that dynamically switch between chain-of-thought and direct responses based on task complexity.

**Knowledge Distillation**: Hinton et al. (2015) introduced the foundational concept of knowledge distillation, where a smaller "student" model learns to mimic the behavior of a larger "teacher" model. Sanh et al. (2019) pioneered the application of knowledge distillation to transformer-based language models, creating a distilled version of BERT Devlin et al. (2019) that retained 97% of BERT's performance while being 60% smaller and 60% faster. More recently, methods such as Jiang et al. (2023); Gu et al. (2023); Liu et al. (2024d); Tian et al. (2025) demonstrate that knowledge distillation can be effectively applied to compress large language models while maintaining their instruction-following capabilities and reasoning performance.

**Chain-of-Thought**:Wei et al. (2022) introduced the concept of chain-of-thought (CoT) prompting, where large language models are encouraged to generate intermediate reasoning steps before its final answer, Kojima et al. (2022) extended chain-of-thought with "step-by-step", revealing that the reasoning capabilities are inherently present in large language models and can be activated through appropriate prompting strategies. Recent work has successfully combined knowledge distillation with chain-of-thought reasoning. Hsieh et al. (2023); Deng et al. (2023); Li et al. (2023) trained student models to generate both intermediate reasoning steps and final answers, demonstrating effective transfer of reasoning capabilities to smaller models with superior performance.

## 3 METHODOLOGY

Our proposed method LLMAD-mini, employs knowledge distillation with our novel hierarchical chain-of-thought to transfer reasoning from teacher LLMs to compact student models. We prompt GPT-4 to analyze log sequences and generate hierarchical reasoning traces, then fine-tune our small model using a multi-task objective combining anomaly classification loss and reasoning alignment loss. During inference, LLMAD-mini outputs both anomaly predictions and structured reasoning traces following the learned format, enabling automated detection with human-interpretable diagnostics as demonstrated in Section 4.

### 3.1 PROBLEM FORMULATION

Given a log sequence which consists of $n$ individual events ordered by timestamp, $E_i = \{e_1, e_2, ..., e_n\}$, generated by a system during execution. Each event $e_i$ represents a structured or semi-structured log entry containing message content describing the occurred event. Our objective is to develop a model $f_\theta : E \rightarrow (R, Y, S)$ that performs interpretable anomaly detection with the following outputs $(R, Y, S)$ where $R = \{r_1, r_2, ..., rk\}$ denotes a Chain-of-Thought reasoning trace comprising k intermediate reasoning steps that progressively analyze the log sequence. The k steps are comprised of 4-level reasonings which are shown in Figure 1.b. $Y \in \{Not\ anomaly, Anomaly\}$ represents the binary anomaly classification, which are natural language instead of numerical values. $S$ provides a context-dependent summary conditioned on $Y$: when $Y = Anomaly$, $S$ contains a root cause analysis identifying the probable failure source and affected components; when $Y = Not\ anomaly$, $S$ is a concise description of the normal system behavior observed about the log sequence.

This formulation enables $f_\theta$ to not only classify anomalies but also provide interpretable explanations through explicit reasoning steps, addressing the critical need for transparency in automated log analysis systems.

## 3.2 FRAMEWORK OVERVIEW

Figure 1.a illustrates LLMAD-mini's teacher-student distillation architecture with three components: (1) a frozen teacher LLM that generates Chain-of-Thought reasoning from input log sequences $E_i = \{e_1, e_2, ..., e_n\}$ and anomaly types; (2) knowledge distillation integrated into fine-tuning process for capability transfer; and (3) a student LLM with LoRA blocks for parameter-efficient learning while preserving pre-trained knowledge.

The teacher model remains frozen during the distillation process, leveraging its pre-trained knowledge to analyze log events and produce structured reasoning outputs. As shown in the figure, the teacher processes various log event types (e.g., instance lifecycle events, VM operations, resource allocations) and generates comprehensive CoT reasoning that identifies critical anomalies such as synchronization failures between control plane and hypervisor components. The student model, equipped with interleaved LoRA adapter layers and decoder blocks (Figure 1.b), undergoes fine-tuning exclusively on the adapter parameters while keeping the base model weights frozen. This architecture enables the student to acquire domain-specific log analysis capabilities without catastrophic forgetting of general language understanding, ultimately producing both CoT reasoning traces and contextual log summaries that indicate whether anomalies are present and their potential root causes.

## 3.3 HIERARCHICAL CHAIN-OF-THOUGHT

Traditional Chain-of-Thought assumes a linear reasoning pathway from input to output Wei et al. (2022), which is insufficient for log anomaly detection by our observation where the relationship between log sequences and root causes is inherently non-sequential. Log anomalies often manifest through complex interactions: a critical failure may result from the confluence of seemingly unrelated events across different components, while the temporal ordering of symptoms may not reflect the actual causal structure. For instance, a memory leak in one service might trigger cascading failures hours later in dependent services, with the true root cause buried among normal operational logs. Moreover, different types of anomalies require different analytical lenses such as some require pattern level analysis of event sequences, while others need detailed inspection of individual event semantics.

To address these challenges, we design a hierarchical Chain-of-Thought structure that decomposes the reasoning process into four progressive stages, as illustrated in Figure 1.b. The hierarchical reasoning begins with two parallel traces event-wise CoT, where the teacher model analyzes individual log events $e_i$ to extract local features such as event severity, component identifiers, and immediate state changes, and stage-wise CoT, which aggregates related events into logical stages representing distinct phases of system operation (e.g., initialization, execution, termination). Then the two paths are subsequently passed into two specialized branches: pattern CoT and indicator CoT. Pattern CoT identifies recurring sequences and temporal patterns across the log sequence, detecting deviations from expected behaviors. Simultaneously, indicator CoT extracts critical signals and anomaly indicators, such as error keywords, performance degradations, or resource exhaustion markers. These parallel reasoning paths capture complementary aspects of the log sequence, pattern CoT focuses on structural anomalies in event ordering and timing, while Indicator CoT targets semantic anomalies in message content and system states.

The hierarchical reasoning culminates in a Final Summary that synthesizes insights from all previous stages. This summary integrates the multiple level observations to produce the final output tuple $(R, Y, S)$, where the reasoning trace $R$ preserves the hierarchical structure, enabling traceable explanations from high level conclusions down to specific event level observations. This hierarchical approach ensures that the student model learns to perform systematic log analysis rather than making superficial pattern matches, as demonstrated in Section 4, we also show the case study in Appendix of figure 2.

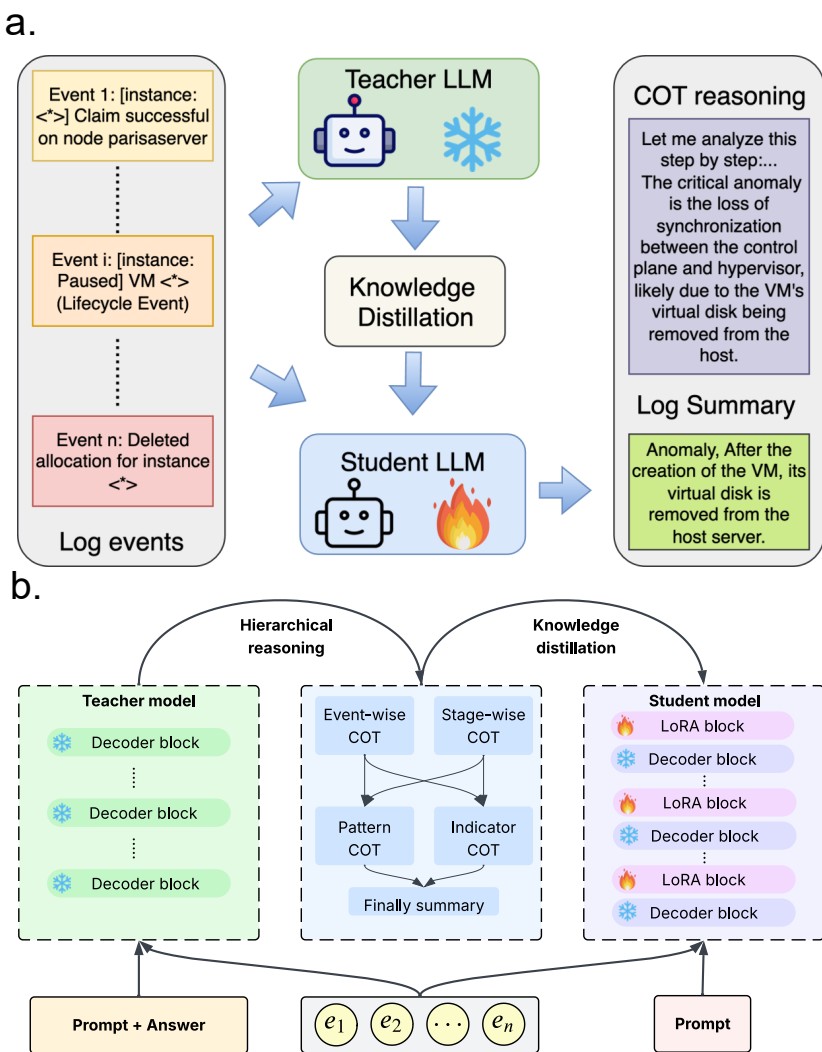

Figure 1: The architecture of LLMAD-mini. a. The overview of LLMAD-mini, consisting of two main components, a frozen teacher model and a distilled student model. b. The framework of teacher model and student model, with the illustration about hierachical CoT reasoning.

## 3.4 KNOWLEDGE DISTILLATION

Our knowledge distillation framework transfers the hierarchical reasoning capabilities from the teacher LLM to a compact student model through a carefully designed training process. As shown in Figure 1.a, the teacher model first processes log sequences with prompts containing few shot examples to generate high quality hierarchical CoT reasoning traces in terms of anomaly classifications. For each training sequence, the teacher produces structured outputs following the hierarchical reasoning pattern described in the previous section. The student model architecture, illustrated in Figure 1.b(right panel), integrates LoRA (Low Rank Adaptation)Hu et al. (2022) blocks with the frozen decoder blocks of a base language model. The LoRA blocks, inserted at regular intervals throughout the network, introduce trainable low rank matrice. During distillation, only the LoRA parameters $\Delta W = BA$ where( $B \in \mathbb{R}^{d \times r}$ and $A \in \mathbb{R}^{r \times k}$ are updated.

we train the student model $f_\theta$ in a way where the distillation objective combines multiple loss components to ensure comprehensive knowledge transfer:

$$\mathcal{L}_{total} = \lambda_1 \mathcal{L}_{CE}(Y_p, Y_t) + \lambda_2 \mathcal{L}_{reason}(R_p, R_t) + \lambda_3 \mathcal{L}_{summary}(S_p, S_t) \tag{1}$$

where $\mathcal{L}_{CE}(Y_p, Y_t)$ ensures accurate student's anomaly classification with ground-truth labels. $\mathcal{L}_{reason}(R_p, R_t)$ minimizes cross entropy loss about student's generated reasoning and teacher's reasoning, while $\mathcal{L}_{summary}(S_p, S_t)$ is mimiking the teacher's summary/root cause identification. $\lambda_1$, $\lambda_2$ and $\lambda_3$ are three parameters that adjusting the weights of the individual losses.Specifically, each $\lambda_i$ is normalized by the token count of its associated output component (classification logits, reasoning traces, and summaries respectively) to ensure balanced gradient contributions regardless of sequence length disparities.

## 4 EXPERIMENTAL RESULTS

### 4.1 EXPERIMENTAL SETUP

We conduct comprehensive experiments to evaluate LLMAD-mini's performance on both in-domain and out-of-domain log anomaly detection tasks. For the student model architecture, we adopt Qwen3-8B Yang et al. (2025) as the base model, which provides an optimal balance between model capacity and computational efficiency. The student model contains approximately 8 billion parameters.

The training process employs supervised fine tuning (SFT) with LoRA adapters configured with rank=16 and $\alpha$=32. We optimize the model using AdamW optimizer with a learning rate of $2e^{-5}$, employing cosine annealing schedule with 10% warmup steps and a weight decay coefficient of 0.02. The total training epochs are 15. Under this configuration, the complete training process requires approximately 2 hours on a single NVIDIA A100 GPU (80GB), while inference can be efficiently performed on a more accessible NVIDIA A10 GPU (24GB).

Our primary dataset is derived from OpenStack cloud infrastructure logs Kalaki et al. (2023), comprising 450 annotated log sequences with balanced representations of normal and anomalous behaviors after filtering. Each sequence contains 10-50 individual log events capturing various system states including VM lifecycle operations, resource allocation activities, and service coordination messages. The dataset exhibits diverse anomaly patterns such as synchronization failures, resource exhaustion, and cascading service failures. We partition the data following an 80/10/10 split for training, validation, and testing, ensuring stratified sampling to maintain anomaly distribution across splits.

To assess generalization capabilities, we evaluate LLMAD-mini on an out-of-domain test set consisting of HDFS (Hadoop Distributed File System) logs Zhu et al. (2023); Jiang et al. (2024). This dataset presents distinct challenges with different log formats, vocabulary, and anomaly patterns compared to OpenStack, providing a rigorous test of the model's transfer learning abilities. The HDFS test set contains 200 sequences with anomalies including block corruption, namenode failures, and datanode disconnections, enabling us to evaluate whether the hierarchical reasoning learned from OpenStack logs generalizes to fundamentally different distributed systems.

### 4.2 BASELINES

We compare LLMAD-mini against two categories of baseline methods to comprehensively evaluate both its anomaly detection performance and reasoning capabilities.

**Traditional Log Anomaly Detection Methods**: We evaluate against state-of-the-art specialized log anomaly detection approaches that represent different methodological paradigms: (1) DeepLog Du et al. (2017), which employs LSTM networks to model log sequences as time series data and detects anomalies through prediction errors; (2) LogAnomaly Meng et al. (2019), which enhances sequence modeling by incorporating semantic information through template embeddings extracted via log parsing; (3) LogBERT Guo et al. (2021), which adapts the BERT architecture specifically for log data by introducing masked log message prediction and hypersphere embedding for anomaly detection; and (4) FastLogAD Lin et al. (2024), which leverages generative adversarial networks (GANs) to learn normal log distributions and identify anomalies through reconstruction errors. These methods represent the evolution of log anomaly detection from sequential modeling to transformer based and generative approaches.

**Large Language Models**: To assess the effectiveness of our knowledge distillation approach, we compare against general purpose LLMs across different scales: (1) Qwen3-8B (base) Yang et al. (2025), our student model architecture without fine tuning, serving as the ablation baseline; (2) Qwen3-32BYang et al. (2025), a larger variant from the same model family to evaluate scaling effects; (3) Llama-3.3-70B-Instruct Dubey et al. (2024), representing the current generation of instruction-tuned models; (4) DeepSeek-V2 Liu et al. (2024a), a mixture-of-experts model optimized for reasoning tasks; and (5) Qwen3-235B-A22BYang et al. (2025), among the largest publicly available models. For fair comparison, all LLMs are evaluated using the same prompt template, but without access to the hierarchical CoT training data.

## 4.3 EVALUATION ON LOG ANOMALY DETECTION

Table 1 presents the comparative results on the OpenStack test set, where we evaluate models using four standard metrics: Precision, Recall, F1-score, and Accuracy. LLMAD-mini achieves superior performance across all metrics, with an F1-score of 0.97, surpassing both traditional methods and general-purpose LLMs. Notably, our 8b-parameter model outperforms models up to 30× larger, demonstrating the effectiveness of domain specific knowledge distillation. Several traditional methods (DeepLog, LogAnomaly) exhibit recall values of 1.0, which our analysis reveals stems from their tendency to classify all sequences as anomalous—a critical failure mode in practical deployments where false positives incur significant operational costs. In contrast, LLMAD-mini maintains balanced precision 1.0 and recall 0.95, indicating robust discrimination between normal and anomalous patterns. The performance gap between the base Qwen3-8B (F1: 0.68) and LLMAD-mini (F1: 0.97) quantifies the contribution of our hierarchical CoT distillation, showing a 42.6% improvement solely from the knowledge transfer process. Furthermore, even the largest baseline LLM (Qwen3-235B-A22B with F1: 0.80) underperforms our compact model, validating that targeted fine-tuning with structured reasoning surpasses raw model scale for specialized tasks like log analysis.

Table 1: Performance on traditional log anomaly detection task

| Methods | Anomaly detection | | | |
| --- | --- | --- | --- | --- |
| | Accuracy | Precision | Recall | F1-Score |
| DeepLog Du et al. (2017) | 0.40 | 0.40 | **1.0** | 0.57 |
| LogAnomaly Meng et al. (2019) | 0.77 | 0.67 | 0.84 | 0.74 |
| LogBert Guo et al. (2021) | 0.80 | 0.83 | 0.78 | 0.81 |
| FastLogAD Lin et al. (2024) | 0.94 | 0.9 | 0.94 | 0.92 |
| Qwen3-8B Yang et al. (2025) | 0.61 | 0.51 | **1.0** | 0.68 |
| Qwen3-32B Yang et al. (2025) | 0.66 | 0.55 | 0.94 | 0.69 |
| Llama3.3-70B-Instruct Dubey et al. (2024) | 0.62 | 0.52 | 0.89 | 0.65 |
| DeepSeek-V2 Liu et al. (2024b) | 0.77 | 0.83 | 0.53 | 0.65 |
| Qwen3-235B-A22B Yang et al. (2025) | 0.81 | 0.69 | 0.95 | 0.80 |
| LLMAD-mini | **0.98** | **1.0** | 0.95 | **0.97** |

## 4.4 PERFORMANCE ON REASONING AND SUMMARIZATION

Beyond binary anomaly classification, we evaluate LLMAD-mini's capability to generate interpretable explanations—a critical requirement for production deployments where operators need actionable insights rather than binary predictions. Tables 2 and 3 present comprehensive evaluations of reasoning quality using standard text generation metrics: Bleu-4, Rouge-1, Rouge-2, and Rouge-L, which measure n-gram overlap between generated and reference explanations at different level.

Table 2 evaluates the quality of hierarchical Chain-of-Thought reasoning traces, where models are supposed to indicate the step-by-step analysis process from individual events to final conclusions. LLMAD-mini achieves a Bleu-4 score of 0.51, representing a 3.2× improvement over the best performing baseline (Qwen3-235B-A22B at 0.16), despite being 30× smaller in parameters. The substantial gains in Rouge-2 (0.46 vs. 0.11, a 4.2× improvement) indicate that our model accurately captures bigram patterns characteristic of technical log analysis, suggesting successful transfer of the teacher's reasoning structure. Notably, general purpose LLMs struggle significantly with this

task, with most achieving Bleu-4 scores below 0.15, demonstrating that the raw model even with larger parameter scale, still cannot compensate for the lack of domain specific reasoning patterns.

Table 2: Performance on CoT reasoning quality

| Methods | CoT Reasoning | | | |
| --- | --- | --- | --- | --- |
| | Bleu-4 | Rouge-1 | Rouge-2 | Rouge-L |
| Qwen3-8B Yang et al. (2025) | 0.11 | 0.25 | 0.08 | 0.11 |
| Qwen3-32B Yang et al. (2025) | 0.14 | 0.32 | 0.09 | 0.13 |
| Llama3.3-70B-Instruct Dubey et al. (2024) | 0.10 | 0.38 | 0.08 | 0.15 |
| DeepSeek-V2 Liu et al. (2024b) | 0.07 | 0.25 | 0.05 | 0.10 |
| Qwen3-235B-A22B Yang et al. (2025) | 0.16 | 0.40 | 0.11 | 0.17 |
| LLMAD-mini | **0.51(3.2×)** | **0.68(1.7×)** | **0.46(4.2×)** | **0.52(3.0×)** |

Table 3 presents results for root cause analysis and log summarization—the final outputs that directly impact operational decision making. Here, the performance gap becomes even more severe: LLMAD-mini achieves a Bleu-4 score of 0.82 compared to 0.05 for Llama-3.3-70B-Instruct, representing a 16.4× improvement. The exceptional Rouge-2 performance (0.82 vs. 0.02, a 41.0× improvement) demonstrates that our model generates root cause explanations with remarkably high fidelity to expert annotations, correctly identifying failures. This dramatic improvement validates that the hierarchical reasoning structure enables the model to synthesize complex observations into accurate, concise diagnoses.

Table 3: Performance on root cause diagnosis/log summary quality

| Methods | Root cause analysis/Log summary | | | |
| --- | --- | --- | --- | --- |
| | Bleu-4 | Rouge-1 | Rouge-2 | Rouge-L |
| Qwen3-8B Yang et al. (2025) | 0.02 | 0.08 | 0.01 | 0.05 |
| Qwen3-32B Yang et al. (2025) | 0.03 | 0.09 | 0.01 | 0.05 |
| Llama3.3-70B-Instruct Dubey et al. (2024) | 0.05 | 0.09 | 0.01 | 0.08 |
| DeepSeek-V2 Liu et al. (2024b) | 0.02 | 0.08 | 0.01 | 0.05 |
| Qwen3-235B-A22B Yang et al. (2025) | 0.03 | 0.09 | 0.02 | 0.05 |
| LLMAD-mini | **0.82(16.4×)** | **0.85(9.4×)** | **0.82(41.0×)** | **0.85(10.6×)** |

The performance differential between LLMAD-mini and larger LLM models provides us with a fundamental insight: for specialized technical domains, targeted knowledge distillation with structured reasoning supervision far outweighs raw parameter count. While general purpose LLMs possess broad knowledge, they lack the specific reasoning patterns required to trace causality through complex system logs. Our distillation process effectively compresses not just the teacher's knowledge but its analytical methodology, enabling a compact 8B parameter model to generate explanations that surpass those from models with up to 235B parameters. This efficiency performance trade-off is particularly valuable for production environments where computational resources are constrained but interpretability requirements are essential.

### 4.5 GENERALIZABILITY ON OUT-OF-DOMAIN LOGS

A critical challenge for log anomaly detection systems is their ability to generalize beyond their training domain, as production environments often encompass heterogeneous systems with diverse logging formats and failure patterns. Despite being fine tuned exclusively on OpenStack logs, we hypothesize that LLMAD-mini's hierarchical reasoning structure enables it to capture fundamental anomaly patterns that transcend specific system implementations. To rigorously evaluate this cross domain transfer capability, we assess performance on HDFS (Hadoop Distributed File System) logs Zhu et al. (2023); Jiang et al. (2024), which exhibit substantially different vocabulary, event types, and architectural patterns compared to OpenStack's cloud infrastructure logs.

Table 4 presents anomaly detection results on the HDFS test set, where LLMAD-mini achieves the highest F1-score of 0.72 and accuracy of 0.70, demonstrating robust generalization despite never encountering HDFS-specific patterns during training. While DeepSeek-V2 achieves higher precision

(0.87), it suffers from poor recall (0.40), indicating overly conservative predictions that miss numerous anomalies. In contrast, LLMAD-mini maintains balanced performance with precision of 0.69 and recall of 0.74, crucial for practical deployment where both false positives and false negatives carry operational costs. Traditional methods (DeepLog, LogAnomaly, LogBERT) fail entirely on this task due to out-of-vocabulary errors, highlighting their inability to handle unseen log templates and system specific terminology without complete retraining.

Table 4: Performance on out-of-domain data about anomaly detection task

| Methods | Anomaly detection | | | |
|---|---|---|---|---|
| | Accuracy | Precision | Recall | F1-Score |
| Qwen3-8B Yang et al. (2025) | 0.5 | 0.5 | 1.0 | 0.67 |
| Qwen3-32B Yang et al. (2025) | 0.56 | 0.54 | **0.88** | 0.67 |
| Llama3.3-70B-Instruct Dubey et al. (2024) | 0.51 | 0.5 | 0.71 | 0.59 |
| DeepSeek-V2 Liu et al. (2024b) | 0.67 | **0.87** | 0.4 | 0.55 |
| Qwen3-235B-A22B Yang et al. (2025) | 0.58 | 0.55 | 0.82 | 0.67 |
| LLMAD-mini | **0.7** | 0.69 | 0.74 | **0.72** |

The reasoning capabilities on out-of-domain data, shown in Table 4, reveal even more striking advantages. For root cause analysis and log summarization, LLMAD-mini achieves Bleu-4 of 0.49 and Rouge-L of 0.50, representing 16.3× and 10× improvements respectively over the best baseline. General purpose LLMs struggle severely with HDFS reasoning, achieving near zero scores (Bleu-4 ranging from 0.01-0.03), as they lack both domain specific knowledge and the structured reasoning patterns necessary for technical log analysis. The 21.5× improvement in Rouge-2 (0.43 vs. 0.02) particularly highlights LLMAD-mini's ability to correctly identify technical terminology and causal relationships even in unfamiliar system contexts.

Table 5: Performance on out-of-domain data about root cause diagnosis/log summary

| Methods | Root cause analysis/Log summary | | | |
|---|---|---|---|---|
| | Bleu-4 | Rouge-1 | Rouge-2 | Rouge-L |
| Qwen3-8B Liu et al. (2024b) | 0.01 | 0.01 | 0.01 | 0.03 |
| Qwen3-32B Liu et al. (2024b) | 0.01 | 0.07 | 0.01 | 0.04 |
| Llama3.3-70B-Instruct Dubey et al. (2024) | 0.03 | 0.02 | 0.01 | 0.02 |
| DeepSeek-V2 Liu et al. (2024b) | 0.01 | 0.06 | 0.01 | 0.02 |
| Qwen3-235B-A22B Liu et al. (2024b) | 0.01 | 0.06 | 0.02 | 0.05 |
| LLMAD-mini | **0.49(16.3×)** | **0.52(7.4×)** | **0.43(21.5×)** | **0.5(10×)** |

## 5 CONCLUSION

We presented LLMAD-mini, a lightweight framework achieving accurate and interpretable log anomaly detection through knowledge distillation from LLMs. Combining hierarchical Chain-of-Thought reasoning with parameter-efficient LoRA fine tuning, our 8B-parameter model outperforms models up to 30× larger on specialized log analysis tasks.

Our hierarchical CoT structure—decomposing into Event-wise, Stage-wise, Pattern, and Indicator levels—captures non-sequential system anomalies more effectively than linear reasoning. Through distillation, LLMAD-mini achieves state-of-the-art detection performance while generating high-quality explanations with up to 41× improvement over general purpose LLMs. The model maintains over 70% accuracy on out-of-domain HDFS logs despite training only on OpenStack, demonstrating it learns fundamental system principles rather than dataset specific patterns.

Practically, LLMAD-mini requires just 2 hours training on a single A100 GPU and runs on consumer hardware, making advanced log analysis accessible for resource-constrained deployments. Our results show targeted distillation with structured reasoning outperforms raw model scaling for specialized domains. Future directions include extending to multi-modal observability data, exploring continual learning for evolving systems, and automated prompt optimization. As distributed systems grow complex, LLMAD-mini advances practical, trustworthy AI powered monitoring—providing necessary interpretability without massive computational overhead.

ETHICS STATEMENT

We acknowledge and adhere to the ICLR Code of Ethics. Our research on log anomaly detection raises no significant ethical concerns. The datasets used (OpenStack and HDFS logs) are publicly available and contain no personally identifiable information, having been collected from system infrastructure rather than user activities. Our method aims to improve system reliability and reduce operational overhead, with no identified potential for harmful applications. The interpretability features of LLMAD-mini promote transparency and accountability in automated system monitoring, allowing operators to understand and verify detection decisions. We have no conflicts of interest to declare, and all computational experiments were conducted using institutional resources in compliance with usage policies. The knowledge distillation approach reduces computational requirements compared to large model deployment, contributing to more sustainable AI practices.

REPRODUCIBILITY STATEMENT

To ensure reproducibility of our work, we provide comprehensive implementation details and resources. The complete model architecture and hyperparameters are specified in section 4.1 and A.2. We will release our code implementation. The OpenStack and HDFS datasets used are publicly available from LogHub, with our preprocessing steps described in Appendix.

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

## A APPENDIX

### A.1 DATASET CURATION

We carefully curate the OpenStack dataset to create high quality training data suitable for hierarchical reasoning. Our preprocessing pipeline consists of several critical steps to ensure data quality and consistency. First, we extract only the essential fields from raw log files, retaining the log content and timestamp columns while removing extraneous metadata such as log levels, source files, and thread identifiers that could introduce noise. For OpenStack logs, we use instance IDs as the primary grouping identifier, aggregating all related log events for each instance into coherent sequences. These sequences are chronologically ordered by timestamp to preserve the temporal dependencies crucial for anomaly detection.

To ensure data quality, we deleted duplicats and filter out incomplete sequences by removing all instance groups that lack complete lifecycle coverage. Specifically, we retain only instances with full event traces from initialization through termination, as partial sequences could mislead the

model during training. This filtering step eliminates approximately 80% of raw data where most of them are excluded due to duplicates but significantly improves training stability and model performance. We then apply log parsing to standardize the format and reduce vocabulary complexity. Variable components such as file paths (e.g., /var/log/nova/compute.log), instance UUIDs (e.g., i-8c7d6e5f4a3b2c1d9e0f), IP addresses, and other long identifiers are replaced with wildcards (*). This abstraction allows the model to focus on log patterns and semantics rather than memorizing specific values. The final curated dataset contains 450 high-quality log sequences for OpenStack, with each sequence comprising 10-50 chronologically ordered events representing complete instance lifecycles. Similar preprocessing is applied to HDFS logs for out-of-domain evaluation, adapting the grouping strategy to use block IDs and datanode identifiers as appropriate for the distributed file system context.

## A.2 TRAINING DETAILS

We implemented LLMAD-mini using the LlamaFactory framework Zheng et al. (2024) for efficient fine-tuning. The training employs supervised fine tuning (SFT) with LoRA adaptation on all model layers, configured with rank 16, alpha 32, and dropout rate of 0.1 for regularization. We use the Qwen model template with a maximum context length of 8,192 tokens to accommodate long log sequences with hierarchical reasoning traces. The optimization process uses AdamW with other settings claimed at section 4.1. We utilize bf16 mixed precision training to reduce memory consumption while maintaining numerical stability.

## A.3 CASE STUDY

We present a detailed case study in Figure 2 demonstrating LLMAD-mini's reasoning process on a real OpenStack VM lifecycle anomaly. Table 6 shows the root cause diagnosis comparison of different baselines on this example.

Table 6: Comparison of predicted anomaly type and root causes for the case study

| Model | Detected root cause |
|---|---|
| LLMAD-mini | **Correct:** "Anomaly, After the creation of the VM, its virtual disk is removed from the host server." |
| DeepLog | **Incorrect:** "Normal"(no root cause) |
| LogBERT | **Partial:** "Abnormal" (no root cause) |
| Llama-3.3-70B-Instruct | **Vague:** "Instance launch failed due to libvirt error and successful cleanup of resources." |
| Deepseek-V2 | **Incorrect:** "The events indicate normal operations such as claiming instances, creating images, plugging and unplugging virtual interfaces, starting, pausing, and stopping instances, deleting instance files, destroying instances on the hypervisor, deallocating networks, and deleting allocations. However, there are also errors such as unexpected events, timeouts, and failed spawning of instances." |
| Qwen3-235B-A22B | **Generic:** "Instance creation failed during libvirt guest startup phase, triggering resource cleanup and termination. Root cause: LibvirtError preventing VM initialization" |

## A.4 THE USE OF LARGE LANGUAGE MODEL

In accordance with ICLR policy, we disclose that Large Language Models were used as a general purpose writing assistant for this manuscript. The LLM's role was strictly limited to language polishing: improving paragraph structure, grammar, and clarity while condensing verbose sections to meet page limits. All technical content, research ideas, methodology, experimental design, and scientific conclusions are entirely original work by the authors. The LLM did not contribute to research ideation or any scientific aspects of the work, serving only as a writing aid similar to grammar checking tools.

**Input Log Sequence:**

```
Event 1: [instance: <*>] Claim successful on node parisaserver
Event 2: [instance: Creating image] <*> <*>
Event 3: Successfully plugged vif <*>
Event 4: [instance: <*>] Failed to start libvirt guest:
libvirt.libvirtError: internal error: process exited
while connecting to monitor
Event 5: Successfully unplugged vif <*>
Event 6: [instance: <*>] Deleting instance files <*>
Event 7: [instance: <*>] Deletion of <*> complete
Event 8: [instance: Terminating instance] <*> <*>
Event 9: [instance: destroyed] Instance <*> successfully.
Event 10: Successfully unplugged vif <*>
Event 11: [instance: <*>] Deletion of <*> complete
Event 12: [instance: destroy] Took <*> seconds to <*>
the instance on the hypervisor.
Event 13: Successfully unplugged vif <*>
Event 14: [instance: <*>] Took <*> seconds to
deallocate network for instance.
Event 15: Deleted allocation for instance <*>
Event 16: [instance: (spawning).] During sync_power_state
the instance has a pending task <*> Skip.
```

**LLMAD-mini Hierarchical Reasoning:**

Event-wise CoT:

- Events 1-3: The VM is successfully claimed on the target node, image creation begins, and the virtual interface (VIF) is plugged in, indicating standard provisioning steps

- Event 4: The attempt to start the VM fails with a 'libvirt.libvirtError: internal error: process exited while connecting to monitor', which is a critical failure during the initialization of the guest VM.

- Events 5-7: Following the start failure, the system begins cleanup: unplugging the VIF, deleting instance files, and confirming deletion, as the VM could not be started.

- Events 8-12: The instance enters termination, is marked as destroyed, and VIFs are unplugged again. The system logs take time to destroy the instance on the hypervisor and deallocate network resources.

- Events 13-15: Additional VIF unplug events and network deallocation confirm the complete cleanup of resources associated with the failed instance.

- Event 16: Power state sync notes pending task.

Stage-wise CoT:

- Normal VM creation (Claim → Image creation → VIF plugged): Events 1-3

- VM start fails (libvirt error): Event 4

- Cleanup initiated (VIF unplugged → Instance files deleted → Deletion complete): Events 5-7

- Instance termination and resource deallocation: Events 8-12

- Final cleanup and allocation deletion: Events 13-16

Pattern CoT: The log sequence follows a typical VM lifecycle up to the point of failed VM start. After the failure, the system transitions directly to cleanup and resource deallocation, with no attempt to recover or retry the failed operation. The presence of multiple VIF unplug and deletion events confirms forced cleanup.

Indicator CoT: **Event 4:** "Failed to start libvirt guest: libvirt.libvirtError: internal error: process exited while connecting to monitor" — critical failure at VM start.
**Events 5-7, 8-12, 13-15:** Multiple VIF unplug, file deletion, and network deallocation events — forced cleanup after failure.
**Event 16:** "During sync_power_state the instance has a pending task" — system acknowledges the anomaly but does not retry.

**Final Diagnosis:** Root Cause: The VM creation process fails at the start phase due to a libvirt internal error, resulting in immediate cleanup and destruction of the instance. The anomaly is detected and handled as a normal destruction event, with no evidence of retries or recovery attempts. Anomaly, After the creation of the VM, its virtual disk is removed from the host server.

Figure 2: Case study showing LLMAD-mini's hierarchical reasoning on a VM lifecycle anomaly

