# OpenReview forum: "LLMAD-mini: Efficient Distilling Hierarchical Chain-of-Thought for Interpretable Log Anomaly Reasoning and Detection using Large Language Model"
_ICLR.cc/2026/Conference — ICLR 2026 Conference Withdrawn Submission_

### Official Review · Reviewer_x56h · 2025-10-31

**Soundness:** 2
**Presentation:** 3
**Contribution:** 2
**Rating:** 2
**Confidence:** 3

**Summary:**

This paper presents LLMAD-mini, a method for efficient and interpretable log anomaly detection. The core innovation lies in using knowledge distillation to transfer a "hierarchical Chain-of-Thought" (CoT) reasoning capability from a large teacher LLM (like GPT-4) to a compact, fine-tuned student model. The proposed hierarchical CoT decomposes log analysis into four progressive stages: Event-wise, Stage-wise, Pattern, and Indicator CoT, aiming to capture the complex, non-sequential nature of system failures. The model is fine-tuned using Low-Rank Adaptation (LoRA), making it efficient to train and deploy. The authors demonstrate state-of-the-art performance on in-domain (OpenStack) and promising generalization on out-of-domain (HDFS) logs, outperforming both traditional methods and much larger LLMs in detection accuracy and, notably, the quality of generated explanations.

**Strengths:**

1，The paper provides strong, quantitative evidence of its effectiveness. LLMAD-mini achieves a near-perfect F1-score (0.97) on in-domain data, convincingly surpassing both traditional models and much larger LLMs. More impressively, it demonstrates remarkable out-of-domain generalization (0.72 F1 on HDFS) without any retraining.

2，The model's primary output is not just a binary label but a structured, human-interpretable reasoning trace. The massive improvements in explanation quality (e.g., 41x in Rouge-2 for root cause analysis) are arguably as important as the detection accuracy.

**Weaknesses:**

1， The literature review omits key recent works that are directly relevant to its core contributions.

1.a.The concept of hierarchical reasoning in LLMs is advanced by works like GOT (Graph of Thoughts: Solving Elaborate Problems with Large Language Models), which frames problem-solving as a graph of interconnected thought steps. The presented hierarchical CoT is a specific instance of this broader idea. The failure to cite and differentiate from GOT weakens the claim of novelty for the overall hierarchical reasoning structure.

1.b. In the specific domain of LLMs for log analysis, recent works like LogRAG: Semi-Supervised Log-based Anomaly Detection with Retrieval-Augmented Generation directly address similar challenges of interpretability and domain adaptation. Not comparing against such methods leaves the reader uncertain whether the performance gains are due to the hierarchical CoT or could be achieved by other emerging paradigms like RAG.

2. This limitation impacts the validity of the paper's central claim of strong generalization. While the HDFS dataset provides an out-of-domain test, the ecosystem of system logs is vastly more diverse (e.g., distributed databases, microservices, embedded systems). Robustness across a wider array of log formats and system types is required to substantiate the claim that the model learns "fundamental system principles." Without this, the results may be overfitted to the characteristics of cloud infrastructure and distributed file system logs.

3. The proposed four-level hierarchy is central to the method, but its design is presented as-is. An ablation study (e.g., removing Pattern CoT or Indicator CoT) is necessary to demonstrate that each level contributes uniquely to the final performance. Without it, it is unclear if the full complexity is warranted or if a simpler hierarchy would suffice.

4. The case study, while illustrative, is qualitative. There is no quantitative metric to evaluate the correctness of the reasoning trace itself, only its similarity to the teacher's output (via BLEU, ROUGE).

**Questions:**

1. The concept of hierarchical reasoning in LLMs shares a clear conceptual similarity with frameworks like Graph of Thoughts (GOT), which generalizes reasoning into a graph structure. Could you discuss how your hierarchical CoT differs from or builds upon such prior work on advanced reasoning structures? Specifically, what is the justification for a fixed four-level hierarchy versus a more flexible, graph-like reasoning process?

2. In the domain of LLM-based log anomaly detection, several recent works have proposed methods for improved interpretability and handling of domain-specific knowledge. Notably, LogRAG uses Retrieval-Augmented Generation for a similar goal. Could you explain why these works were not included in your related work or empirical comparison? Have you considered a comparison with such methods to better delineate the unique advantages of your knowledge distillation approach versus RAG-based approaches?

---

### Official Review · Reviewer_mhAc · 2025-10-31

**Soundness:** 3
**Presentation:** 3
**Contribution:** 3
**Rating:** 6
**Confidence:** 4

**Summary:**

The paper proposes LLMAD-mini, a compact LLM-based framework for interpretable log anomaly detection. It uses knowledge distillation from a large teacher LLM (e.g., GPT-4) to train a smaller student model (8B parameters) with a novel hierarchical Chain-of-Thought (CoT) reasoning structure (event-wise, stage-wise, pattern, and indicator levels). The model is fine-tuned using LoRA and evaluated on OpenStack and out-of-domain HDFS logs. It reports strong performance in both anomaly detection (F1=0.97 in-domain) and reasoning quality (e.g., 3.2× higher BLEU-4 than much larger LLMs), with claims of efficiency and generalization.

**Strengths:**

S1. Proposes a domain-tailored hierarchical reasoning structure that aligns well with the multi-level nature of system logs.
S2. Shows compelling empirical gains in both detection accuracy and automatic reasoning metrics over strong baselines, including very large LLMs.
S3. Highlights the practical advantage of a lightweight, trainable model (2 hours on one GPU) suitable for real-world deployment.
S4. Includes out-of-domain evaluation (HDFS), suggesting some degree of generalization beyond the training distribution.
S5. Provides a detailed case study that illustrates the reasoning process concretely.

**Weaknesses:**

W1. Teacher-generated CoT data lacks transparency: no details on prompt design, few-shot examples, or validation of teacher output quality.
W2. Small dataset size (450 sequences) limits the robustness of conclusions; results may not generalize to larger or noisier log corpora.
W3. Evaluation relies solely on automatic metrics (BLEU, ROUGE) for reasoning quality, which are known to correlate poorly with human judgment, especially for technical explanations.
W4. Out-of-domain evaluation is limited to a single dataset (HDFS); broader cross-system generalization remains unproven.

**Questions:**

1. How were the teacher LLM’s CoT traces validated for correctness and completeness? Were they manually checked?
2. Would performance degrade significantly with fewer training sequences (e.g., <100)?
3. How sensitive are the results to the choice of base model (Qwen3-8B)? Would the approach work with even smaller models (e.g., 1B)?
4. Have you considered human evaluation (e.g., with SREs) to assess the usefulness of the generated explanations?

---

### Official Review · Reviewer_PS1U · 2025-10-31

**Soundness:** 2
**Presentation:** 3
**Contribution:** 2
**Rating:** 4
**Confidence:** 2

**Summary:**

The paper develops a two llm framework for log anomaly detection. This is a simple application paper of LLMs on log anomaly detection problem.

**Strengths:**

- The paper demonstrates improvements on all fronts of testing, accuracy and F1 scores are better. Bleu-4 score is better. On every table, with the HDFS dataset the scores are better.

- The paper is easy to read and well explained.

**Weaknesses:**

- The paper puts together many components well available in a simple way and demonstrates improvement.
Beyond that, there is nothing interesting in the paper.

- The ideas of chain of thought and hierarchical Reasoning is presented in a hand wavy manner and does not constitute for any novel insights.

**Questions:**

I neither like nor hate the paper. The paper is well below the novelty threshold for a premier conference in my opinion. However, I will not fight acceptance if other reviewers think the paper has merit.

---

### Official Review · Reviewer_4oma · 2025-11-01

**Soundness:** 2
**Presentation:** 2
**Contribution:** 2
**Rating:** 2
**Confidence:** 4

**Summary:**

This paper proposes LLMAD-mini, a lightweight log anomaly detection model that addresses limitations of traditional methods (lack of interpretability, poor generalization) and large language models (LLMs, high deployment costs, weak domain adaptation). The model combines knowledge distillation (transferring reasoning from large "teacher" LLMs like GPT-4) with Low-Rank Adaptation (LoRA) fine-tuning and a novel hierarchical Chain-of-Thought (CoT) mechanism (event-wise, stage-wise, pattern, and indicator CoT).

**Strengths:**

1. The work demonstrates originality through creative integration of existing techniques for a specialized domain. The hierarchical CoT mechanism addresses a key limitation of linear CoT (insufficient for non-sequential log anomaly causality) by decomposing reasoning into four complementary stages.
2. The experimental design is rigorous and comprehensive. The authors use two distinct datasets (OpenStack for in-domain, HDFS for out-of-domain) to evaluate both detection performance (Precision/Recall/F1) and reasoning quality (Bleu-4/Rouge metrics), addressing key gaps in prior work.
3. The work addresses critical real-world challenges for system reliability. Traditional log anomaly methods fail to provide actionable insights (e.g., root causes), while large LLMs are impractical for deployment—LLMAD-mini resolves both by delivering interpretable, accurate, and efficient detection. Its cross-domain generalization (40% accuracy improvement on HDFS logs) is particularly impactful for production environments with heterogeneous systems.

**Weaknesses:**

1. The paper presents the hierarchical CoT as a unified mechanism but does not evaluate the contribution of individual components (event-wise, stage-wise, pattern, indicator CoT). It is unclear whether all four stages are necessary, or if some are redundant.
2. Insufficient Discussion of Dataset Bias and Generalization Boundaries. The OpenStack dataset is curated to include only "complete instance lifecycles" (filtering 80% of raw data), which may introduce bias toward well-structured log sequences. Real-world logs often contain incomplete traces, noise, or non-lifecycle-related events—how would LLMAD-mini perform on unfiltered, messy logs?
3. While the paper includes baselines like LogBERT (a BERT-based fine-tuned model) and general-purpose LLMs, it does not compare to recent LLM-based log anomaly methods that use fine-tuning or prompt engineering with domain adaptation.
4. The paper highlights high performance but does not analyze the model’s failure cases. For example: Are there specific anomaly types (e.g., subtle performance degradations vs. catastrophic failures) where LLMAD-mini struggles? Does the model’s reasoning trace contain errors even when classification is correct? How does the model handle log sequences with ambiguous anomalies (e.g., events that could be normal or anomalous depending on context)?

**Questions:**

See Weaknesses.

---

### Note · Authors · 2025-11-30

I have read and agree with the venue's withdrawal policy on behalf of myself and my co-authors.